

# Joint user grouping and power control using whale optimization algorithm for NOMA uplink systems

Bilal ur Rehman[1], Mohammad Inayatullah Babar[1], Arbab Waheed Ahmad[2], Muhammad Amir[1], Waleed Shahjehan[1], Ali Safaa Sadiq[3], Seyedali Mirjalili[4,6] and Amin Abdollahi Dehkordi[5]

[1] Department of Electrical Engineering, University of Engineering and Technology, Peshawar, Pakistan
[2] Department of Electrical and Computer Engineering, PAF-IAST, Haripur, Pakistan
[3] School of Mathematics and Computer Science, University of Wolverhampton, Wulfruna Street Wolverhampton, WV1 1LY, United Kingdom
[4] Centre of Artificial Intelligence Research and Optimisation, Torrens University, Brisbane, Australia
[5] Computer Engineering Faculty, Najafabad Branch, Islamic Azad University, Najafabad, Iran
[6] Yonsei Frontier Lab, Yonsei University, Seoul, South Korea

Corresponding authors
Bilal ur Rehman,
bur@uetpeshawar.edu.pk
Ali Safaa Sadiq, ali.sadiq@wlv.ac.uk

## ABSTRACT

The non-orthogonal multiple access (NOMA) scheme has proven to be a potential candidate to enhance spectral potency and massive connectivity for 5G wireless networks. To achieve effective system performance, user grouping, power control, and decoding order are considered to be fundamental factors. In this regard, a joint combinatorial problem consisting of user grouping and power control is considered, to obtain high spectral-efficiency for NOMA uplink system with lower computational complexity. To solve the joint problem of power control and user grouping, for Uplink NOMA, we have used a newly developed meta-heuristicnature-inspired optimization algorithm *i.e.*, whale optimization algorithm (WOA), for the first time. Furthermore, for comparison, a recently initiated grey wolf optimizer (GWO) and the well-known particle swarm optimization (PSO) algorithms were applied for the same joint issue. To attain optimal and sub-optimal solutions, a NOMA-based model was used to evaluate the potential of the proposed algorithm. Numerical results validate that proposed WOA outperforms GWO, PSO and existing literature reported for NOMA uplink systems in-terms of spectral performance. In addition, WOA attains improved results in terms of joint user grouping and power control with lower system-complexity when compared to GWO and PSO algorithms. The proposed work is a novel enhancement for 5G uplink applications of NOMA systems.

## INTRODUCTION

Multiple access approaches are increasingly gaining importance in modern mobile communication systems, primarily due to the overwhelming increase in the communication demands at both the user and device level. Over past few years, non-orthogonal multiple access (NOMA) (*Ding et al., 2017a*; *Ding et al., 2014*; *Ding et al., 2017b*; *Benjebbovu et al.,*

*2013*) schemes have earned significant attention for supporting the huge connectivity in contemporary wireless communication systems. The NOMA schemes are currently considered to be the most promising contender for the 5G and beyond 5G (B5G) wireless communications, which are capable of accessing massive user connections and attaining high spectrum performance. Moreover, a report has been published recently regarding the Third Generation Partnership Project for determining the effectiveness of NOMA schemes for several applications or development scenarios, particularly for ultra-reliable low latency communications (URLLC), enhanced mobile broadband (eMBB), and massive machine type communications (mMTC) (*Benjebbour et al., 2013*). Contrary to the classic orthogonal multiple access (OMA) approaches, the NOMA schemes can offer services to multiple users in the same space/code/frequency/time resource block (RB). The NOMA schemes are also capable of differentiating the users that have distinct channel settings. These schemes are mainly inclined at strengthening connectivity and facilitating users with an efficient broad-spectrum (*Islam et al., 2016*; *Dai et al., 2015*).

Some recent studies (*Chen, Wang & Zhang, 2018*; *Wang et al., 2019*; *Shahini & Ansari, 2019*) have discussed the effective use of the NOMA approach in standard frameworks for Internet of Things (IoT) systems and Vehicle-to-Everything (V2X) networks. The successive interference cancellation (SIC) technique, which is pertinent for multi-user detection and decoding is implemented for the NOMA scheme at the receiver end. The SIC technique operates differently for the downlink and uplink scenarios. In the downlink NOMA scenario, SIC is applied at the receiver end, where high energy is consumed during processing when a lot of users are considered in the NOMA group. For that reason, two users are typically considered in a group for optimum grouping/pairing of users in the case of the downlink NOMA system (*Al-Abbasi & So, 2016*; *He, Tang & Che, 2016*). Whereas in the uplink NOMA systems, it is possible to employ SIC at the base station (BS) that has a higher processing capacity. Moreover, in uplink NOMA, multiple users are allowed to transmit in a grant-free approach that leads to a significantly reduced latency rate.

From a practical perspective, the user-pairing/grouping and power control schemes in uplink/downlink NOMA systems are critically required to achieve an appropriate trade-off between the performance of the NOMA system and the computational complexity of the SIC technique. Over the past few years, several studies have discussed different prospects regarding the maximization of sum rate (*Zhang et al., 2016a*; *Ding, Fan & Poor, 2015*; *Ali, Tabassum & Hossain, 2016*), the transmission power control approaches (*Wei et al., 2017*), and fairness (*Liu, Mähönen & Petrova, 2015*; *Liu et al., 2016*) for user pairing/grouping NOMA systems. Regarding the maximization of sum rate, a two-user grouping scheme based on a unique channel gain is demonstrated in *Ding, Fan & Poor (2015)* whereas another study (*Ali, Tabassum & Hossain, 2016*) presented a novel framework for pertinent user-pairing/grouping approaches to assign the same resource block to multiple users.

In reference to the user pairing schemes (*Sedaghat & Müller, 2018*) used the Hungarian algorithm with a modified cost function to investigate optimum allocation for three distinct cases in the uplink NOMA system. Furthermore, several matching game-based (*Liang et al., 2017*) user-pairing/grouping approaches are discussed in *Xu et al. (2017)* and *Di, Song & Li (2016)*, wherein the allocation of users and two sets of players are modeled

as a game theory problem. Numerous recent studies (*Zhai et al., 2019*; *Zhu et al., 2018*; *Nguyen & Le, 2019*) have also investigated different user-pairing/grouping schemes for NOMA systems. A novel algorithm named Ford Fulkerson (*Zhai et al., 2019*) has been introduced for D2D cellular communication to address the user-pairing issue in NOMA systems. In addition to that, optimal user paring is achieved in *Zhu et al. (2018)* by taking two users with appropriate analytical conditions into consideration. A new framework (*Song et al., 2014*) is also presented for optimum cooperative communication networks. Besides that, a lookup table (*Azam, Shahab & Shin, 2019*) is introduced by performing comprehensive calculations to highlight the significance of power allocation and uplink user pairing in obtaining high sum-rate capacity while fulfilling the demands of user data rates. For the uplink case, a cumulative distributive function (CDF)-based resource allocation scheme (*Zhanyang, Toor & Jin, 2018*) is presented where for each time slot, the selection of two users is dependent on the highest value of the CDF. Moreover, a few dynamic power allocation and power back-off schemes are also discussed in few studies (*Zhang et al., 2016b*; *Yang et al., 2016*) for scrutinizing the performance of the system to obtain high sum rates and meet the service quality requirements.

In the context of overlapping, a generic user grouping approach (*Chen et al., 2020a*) is presented for NOMA, which involves the grouping of many users with a limitation on maximum power. The authors also formulated a problem for generalized user grouping and power control to achieve an optimized user grouping scheme based on the machine learning approach. Furthermore, another study (*Chen et al., 2020b*) proposed a framework in which an overlapping coalition formation (OCF) game is used for overlapping user grouping and an OCF-based algorithm is also introduced that facilitated the self-organization of each user in an appropriate overlapping coalition model. Besides that, a joint problem is examined in *Guo et al. (2019)* for user grouping, association, and power allocation in consideration of QoS requirements for enhancing the uplink network capacity. *Zhang et al. (2019)* also discussed a joint combinatorial problem for obtaining a sub-optimal and universal solution for user-pairing/grouping to boost the overall system performance. Additionally, the authors in *Wang et al. (2018)* considered a user association problem by using an orthogonal approach for grouping users and employing a game-theoretic scheme for the allocation of a resource block to multi-users in a network. It has been observed that there are certain limitations associated with the game-theoretic schemes that are typically employed in user association techniques. However, the evolutionary algorithms (EAs) are universal optimizers that exhibit exceptional performance irrespective of the optimization problems being studied. The problem formulation is done as a sum rate utility function for the network and a parameter is presented that depicts the intricacy for power control problems. Therefore, the parameters for power control remain constant for all the systems. Moreover, NOMA-based mobile edge computing (MEC) system (*Zheng, Xu & Tang, 2020*) has been investigated to improve the energy efficiency during task offloading process. Further, a matching coalition scheme has been used to address the issue of power control and resource allocation. In addition, a matching theory (*Panda, 2020*) approach is proposed to enhance the operational system's user patterns and resource management.

Meta-heuristics are high-level processes that combine basic heuristics and procedures in order to provide excellent approximation solutions to computationally complex combinatorial optimization problems in telecommunications (*Martins & Ribeiro, 2006*) . Furthermore, the key ideas connected with various meta-heuristics and provide templates for simple implementations. In addition, several effective meta-heuristic approaches to optimization problems have been investigated in telecommunications.

Several meta-heuristic algorithms (*Sharma & Gupta, 2020*) have been proposed to address localization problems in sensor networks. Some of the meta-heuristic algorithms used to solve the localization problems include the bat algorithm, firework algorithm and cuckoo search algorithm. For wireless sensors networks (*Wang, Li & Pedrycz, 2020*), routing algorithm has been developed based on elite hybrid meta-heuristic optimization algorithm.

On the other hand, swarm intelligence (SI) algorithms, in addition to game theory and convex optimization, has recently emerged as a promising optimization method for wireless-communication. The use of SI algorithms can resolve arising issues in wireless networks such as power control problem, spectrum allocation and network security problems (*Pham et al., 2020b*). Furthermore, two SI algorithms, named grey wolf optimizer (GWO) and particle swarm optimizer (PSO) are also used in literature for solving the joint problem regarding user associations and power control in NOMA downlink systems to attain maximized sum-rate (*Goudos et al., 2020*). Additionally, an efficient meta-heuristic approach known as multi-trial vector-based differential evolution (MTDE) (*Nadimi-Shahraki et al., 2020*) has been implemented for solving different complex engineering problems by using multi trial vector technique (MTV) which integrates several search algorithms in the form of trial vector producers (TVPs) approach. Recently, an updated version of GWO *i.e.,* Improved-grey wolf optimizer (I-GWO) (*Nadimi-Shahraki, Taghian & Mirjalili, 2021*) has been investigated for handling global optimization and engineering design challenges. This modification is intended to address the shortage of population variety, the mismatch between exploitation and exploration, and the GWO algorithm's premature convergence. The I-GWO algorithm derives from a novel mobility approach known as dimension learning-based hunting (DLH) search strategy which was derived from the natural hunting behaviour of wolves. DLH takes a unique method to creating a neighbourhood for each wolf in which nearby information may be exchanged among wolves. This dimension learning when employed in the DLH search technique improves the imbalance between local and global search and preserves variation. A parallel variant of the Cuckoo Search method is the Island-based Cuckoo Search (IBCS) (*Alawad & Abed-alguni, 2021*) using extremely disruptive polynomial mutation (iCSPM). The discrete iCSPM with opposition-based learning approach (DiCSPM) is a version of iCSPM has been proposed to schedule processes in cloud computing systems focusing on data communication expenses and computations. Moreover, for scheduling dependent tasks to virtual machines (VMs), this work offers a discrete variant of the Distributed Grey Wolf Optimizer (DGWO) (*Abed-alguni & Alawad, 2021*). In DGWO, the scheduling process is considered as a problem of minimization for data communication expenses and computation.

In this paper, a joint combinatorial problem of user pairing/grouping, power control, and decoding order are considered for every uplink NOMA user within the network. To solve this problem, we propose a recently introduced meta-heuristic algorithm known as whale optimization algorithm (WOA) (*Mirjalili & Lewis, 2016*) that is inspired by the hunting approach of the humpback whale. Furthermore, a grey wolf optimizer (GWO) (*Mirjalili, Mirjalili & Lewis, 2014*) and particle swarm optimization (PSO) (*Kennedy & Eberhart, 1995*) algorithms are also employed in this research study. The results obtained through the algorithms proposed in *Sedaghat & Müller (2018)*, WOA, GWO and the popular PSO are exclusively compared in this study. The acquired results indicate that the WOA outperformed the existing algorithm (*Sedaghat & Müller, 2018*), GWO and PSO in-terms of spectral-efficiency with lower computational complexity.

The rest of the paper is structured as follows: the 'System Model and Problem Formulation' describes the mathematical representation and research problem of NOMA uplink system. The solution is provided in the 'Solution of Proposed Problem' section where an efficient decoding order, power control scheme and user grouping approach are employed for NOMA uplink System. A concise analysis on the simulation is provided in 'Simulation Results' section. The 'Conclusion' section presents the summary of this research work.

## SYSTEM MODEL AND PROBLEM FORMULATION

### System model

As illustrated in Fig. 1, we consider an uplink NOMA transmission with a single-cell denoted by $C$. The number of users $M$ served by a single base station (BS) placed at the centre of the cell. To obtain the signal/information requirements of several users, the number of physical resource block (PRB) denoted by $N$ are assigned to multiple-users in a cell.

For uplink transmission, users in almost in same PRB/group maintaining same PRB execute NOMA operation while users belong to different PRB/group assigned different PRB execute OMA operation. Hence, the received signal $z_n$ at the BS can be represented as:

$$z_n = \sum_{m=1}^{M} \upsilon_{n,m} \, g_m \sqrt{\alpha_m P} s_m + \omega_n. \tag{1}$$

where $\upsilon_{n,m} \in \{0,1\}$ is the user $n$ indicator assigned to the $n-th$ group. The transmission path between user $m$ and BS is represented by $g_m$ which is Guassian distributed. The power control coefficients is denoted by $\alpha_m (0 \leq \alpha_m \leq 1)$. For each user $m$, the transmission power and the signal is denoted by $P$ and $s_m$, where $\mathbb{E}(|s_m|^2 = 1)$. The additive white Gaussian noise (AWGN) power is denoted by $\omega_n$ with an average power $\sigma^2$. Therefore, the maximum spectral efficiency of user $M$ and the received signal to interference plus noise ratio (SINR) on $n-th$ PRB can be expressed as:

$$S_m = log_2 \; (1 + \phi_m) \tag{2}$$

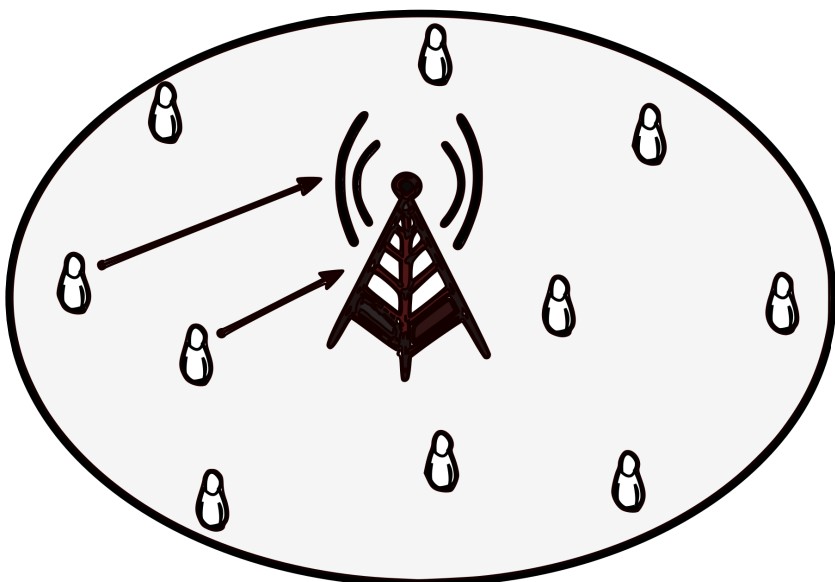

**Figure 1   NOMA uplink transmission.**

$$\phi_m = \frac{|g_m|^2 \alpha_m P}{\sum_{j \neq m}^{M} |g_j|^2 \alpha_j P + \sigma^2} \tag{3}$$

The SIC operation is carried out at the BS for each PRB/group to decode the users signal. The decoding order of a user $n$ is represented by $\delta_{n,m}$ in a cell, where $\delta_{n,m} = a > 0$ assumes that any user $m$ in a group is the $i - th$ one in the $n - th$ PRB is to be decoded. Thus, the maximum spectral efficiency of user $n$ can be represented as:

$$S_m = log_2 \left( 1 + \frac{|g_m|^2 \alpha_m \gamma}{\sum_{\substack{j \neq m \\ \delta_{n,j} > \delta_{n,m} > 0}}^{M} |g_j|^2 \alpha_j \gamma + 1} \right) \tag{4}$$

where $\delta_{n,j} > \delta_{n,m}$ represents the decoding order of users in a PRB/group. If users $m$ and $j$ are in the same group, then it implies that user $m$ is decoded first. The transmission power to noise ratio is represented by $\gamma$, where $\gamma = P/\sigma^2$. Assuming that, channel-state-information (CSI) is known by BS of each user within coverage area.

To attain effective user-pairing/grouping and power control for NOMA uplink system, each user $M$ in a cell transmit their power control coefficient $\alpha_m$ along with user indicator $\upsilon_{n,m}$. Hence, the maximum spectral-efficiency in the $n - th$ PRB/group can be expressed as follows:

$$S_t(n) = \sum_{\upsilon_{n,m}=1} S_m \tag{5}$$

$$S_t(n) = \log_2\left(1 + \sum_{\upsilon_{n,m}=1} |g_m|^2 \alpha_m \gamma\right) \tag{6}$$

Eq. (6) clearly shows that the spectral efficiency in each group has not been affected by the order of decoding but has an impact on each user.

**Problem formulation**

In this paper, we propose an efficient method for power-control, decoding order and user-pairing/grouping to increase the spectral-efficiency under their required minimum rate constraint. Therefore, a joint combinatorial problem of power control, decoding order and user pairing/grouping is formulated to maximize the spectral-efficiency. The minimum spectral-requirement of each user in the network is $s_m$. Therefore, the spectral efficiency maximization problem (*Sedaghat & Müller, 2018*; *Zhang et al., 2019*) can be formulated as:

$$\text{maximize}_{\{\upsilon_{n,m}\},\{\delta_{n,m}\}\in\pi,\{\alpha_m\}} \quad S_t = \sum_{n=1}^{N} S_t(n) \tag{7a}$$

$$\text{subject to} \quad C_1 : 0 \leq \alpha_m \leq 1, \forall m, \tag{7b}$$

$$C_2 : S_m \geq s_m, \forall m, \tag{7c}$$

$$C_3 : \upsilon_{n,m} \in \{0,1\}, \forall m, \forall n, \tag{7d}$$

$$C_4 : \sum_{n=1}^{N} \upsilon_{n,m} = 1, \forall m \tag{7e}$$

where $\delta_{n,m}$ represents the decoding order and $\pi$ indicates all possible combinations of users decoding orders in a network. $C_1$ indicates the upper bound of transmission power. $C_2$ guarantees the minimum rate of a user. $C_3$ and $C_4$ ensures the user indicator and $m$ users assigned to PRB/group.

## SOLUTION OF PROPOSED PROBLEM

To achieve the global optimal solution for problem (Eq. (7a)), the optimization variables $\upsilon_{n,m}, \delta_{n,m}$, and $\alpha_m$ are strongly correlated, which makes the problem complex. In connection of the fact that user-pairing variables $\delta_{n,m}$ are combinatorial integer programming variables. Hence, first solve the combinatorial problem of power control and decoding order instead and compute the optimum user-pairing/grouping solution. In case of any fixed scheme of user-grouping, the value of $\upsilon_{n,m}$ are independent among all distinct group regarding both decoding order and power control.

$$\text{maximize}_{\{\delta_{n,m}\}\in\pi,\{\alpha_m\}} \quad S_t(n) \tag{8a}$$

$$\text{subject to} \quad S_m \geq s_m, \quad m \in \mathcal{M}_n, \tag{8b}$$

$$0 \leq \alpha_m \leq 1, \quad m \in \mathcal{M}_n \tag{8c}$$

where $\mathcal{M}_n$ indicates set of all possible combination in the $n-th$ PRB.

## Optimal decoding for optimal user-pairing/grouping

In order to apply the SIC operation, all users signals/information are decoded by the receiver in the descending order based on channel condition. In the uplink NOMA system (*Ali, Tabassum & Hossain, 2016*), the users with better channel condition is decoded first at the BS while the user with worse channel conditions is decode last. As a result, the user with better channel condition experiences interference from all the users in the network, while the users with poor channel condition experiences interference free transmission.

To attain an efficient decoding (*Zhang et al., 2019*) for NOMA uplink users, the decoding order for $M$ users in a cell concern to same group/PRB, based upon the value of $J_n$, where different decoding order of each user in a network depend on power control (*Zhang et al., 2019*) scheme regarding different feasible region can be represented as:

$$J_m = |g_m|^2 (1 + \frac{1}{\Psi_m}) \tag{9}$$

where

$$\Psi_m = 2^{s_m} - 1 \tag{10}$$

Based on Eq. (9), the user with higher value of $J_m$ in a cell is decoded first. Also applies that the decoding-order does not affect the spectral efficiency of each PRB/group.

## Power control

The nature of the problem in Eq. (8a) is a mixed integer non-linear programming (MINLP). Hence, we have achieved the optimal solution for decoding order $\delta_{n,m}$. Therefore, it is required to find all the possible group of combination for user pairing/grouping.

For this purpose, $k$ users in a single cell $C$ are considered. Without loss of generality, it is needful to reduce the complexity and simplify the mathematical procedure regarding optimal decoding order $\delta_{n,m}$. The users are listed in a $C$ based on the decreasing order $J_m$, for example $1, 2, 3, \ldots, K$. Therefore, Eq. (8a) can be represented as:

$$\text{maximize}_{\{\alpha_k\}} \sum_{k=1}^{K} |g_k|^2 \alpha_k \gamma \tag{11a}$$

$$\text{subject to} \quad |g_k|^2 \alpha_k \gamma \geq \Psi_k \left( \sum_{j=k+1}^{K} |g_j|^2 \alpha_j \gamma + 1 \right), \forall k, \tag{11b}$$

$$0 \leq \alpha_k \leq 1, \forall k, \tag{11c}$$

where $\{\alpha_k\}$ represents power control-variables. Equations (11a) and (11b) show linearity and translated to SNR formulations respectively.

As shown in Eq. (11a), $\alpha_k$ is increasing. Therefore, the optimal solution for power control will always be upper bound. To determine the lower bound of power control (*Zhang et al., 2019*), the following equation can be solved as:

$$\alpha_k^0 = \frac{\Psi_k \gamma_k'}{|g_k|^2 \gamma}, \quad 1 \le k \le K \tag{12}$$

where

$$\gamma_k' = \prod_{u=k+1}^{K} (\Psi_u + 1) \tag{13}$$

which signifies that the spectral efficiency requirements is equal to the sum of spectral efficiencies of all the users. If $\alpha_k^0 \ge 1$, exceeds the limit of upper bound and hence, no feasible solution for Eq. (11a). If $0 \le \alpha_k^0 \le 1$, Eq. (11a) has the feasible solution due to bound of $\alpha_k$ variables. Therefore, for all users $M$ in a cell, the optimal solution (*Zhang et al., 2019*) of the $\alpha_k$ variables can be illustrated as

$$\alpha_k^* = min\{1, b_k\} \tag{14}$$

where

$$b_k = min\{\frac{|h_u|^2 \gamma}{\Psi_u} - \sum_{q=u+1}^{k-1} |h_q|^2 \gamma - \sum_{j=k+1}^{K} |h_j|^2 \alpha_j^0 \gamma - 1(u = 1, 2, 3, \ldots, k-1)\}. \tag{15}$$

In reference to Eqs. (14) and (15), the optimal power control variables $\alpha_k^*$ mentioned in problem Eq. (11a) is achieved. Specifically, if $\alpha_k^* = b_k$, for other users, the optimal power control variables are $\alpha_j^* = \alpha_j^0$ for $j > k$.

## User grouping

An efficient and low computational time algorithm for user-pairing/grouping is one of the key concern for an effective NOMA uplink system. In this regard, three different meta-heuristic algorithms are proposed to solve the issue of complexity. The WOA is investigated for an efficient optimal and sub-optimal solution for user-pairing/grouping problem as a result to enhance the system performance. Further, the user pairing/grouping problem that exploits the channel-gain difference among different users in a network and the objective is to raise system's spectral-efficiency. To determine the optimum user-pairing/grouping, a specific approach of solving user pairing/grouping problem is by using the search approach. For fixed user-pairing/grouping scheme, the optimal solution is obtained (*Zhang et al., 2019*). Then, list all the users in the decreasing order of $J_m$ accordingly. The proposed algorithm for user pairing/grouping problem is illustrated in Algorithm 1. Initially, define the feasible solution of user grouping for exhaustive and swarm based algorithm. An exhaustive search explores each data points within the search region and therefore provides the best available match. Furthermore, a huge proportion of computation is needed. Particularly a discrete type problem where no such solution

exists to find the effective feasible solution. There may be a need to verify each and every possibility sequentially for the purpose of determining the best feasible solution. The optimal solution using exhaustive search algorithm (*Zhang et al., 2019*) is getting obdurate because the number of comparison increases rapidly. Hence, the system complexity of WOA for user grouping scheme is $O(MN)$, where as $O(N^M)$ represent the complexity of the exhaustive search algorithm. Therefore, a WOA approach is employed to reduce the complexity and provide efficient results. In addition, GWO and PSO algorithms are also proposed for the same problem.

## Whale optimization algorithm (WOA)

To enhance the spectral-throughput and reduce the system complexity, an innovative existence meta-heuristic optimization technique named whale optimization algorithm (WOA) (*Mirjalili & Lewis, 2016*) is proposed in this paper. The algorithm WOA is resembles to the behaviour of the humpback whales, which is based on the bubble-net searching approach. Three distinct approaches are used to model the WOA is described as

### Encircling prey

In this approach, the humpback-whales can locate the prey-location of the prey and en-circle that region. Considering that, the location of the optimal design in the search region is not known in the beginning. Hence, the algorithm WOA provides the best solution that is nearer to the optimal value. First determine the best solution regarding location and then change the position according to the current condition of the other search agents concerning to determine the best solution. Such an approach is described mathematically and can be expressed as:

$$\overrightarrow{E} = |A.\overrightarrow{X^*}(t) - X(t)| \tag{16}$$

$$\overrightarrow{Y}(t) = \overrightarrow{X}(t+1) \tag{17}$$

$$\overrightarrow{Y}(t) = \overrightarrow{X^*}(t) - \overrightarrow{B}.\overrightarrow{E} \tag{18}$$

where, $\overrightarrow{B}$ and $\overrightarrow{A}$ represents the coefficients-vectors, $t$ defines the initial iteration and $X^*$ and $\overrightarrow{X}$ both describes the position- vector where $X^*$ includes the best solution so far acquired. $||$ and $.$ defines the absolute and multiplication. Noted that the position vector $X^*$ is updated for each iteration until to find the best solution. The coefficients vector vectors $\overrightarrow{B}$ and $\overrightarrow{A}$ can be determined as:

$$\overrightarrow{B} = 2\overrightarrow{b}.\overrightarrow{r} - \overrightarrow{b} \tag{19}$$

$$\overrightarrow{A} = 2.\overrightarrow{r} \tag{20}$$

where $\overrightarrow{r}$ indicates random vector $0 \leq r \leq 1$ and $\overrightarrow{b}$ represent a vector with a value between 2 and 0, which is decreasing linearly during the iteration.

### Spiral bubble-net feeding maneuver

Two techniques are proposed to predict accurately the bubble-net activity of humpback-whales.

### 1. Shrinking en-circling

This type of techniques is achieved by decreasing the value of $\vec{b}$ using Eq. (19). It is to be noted that the variation range of $\vec{B}$ is also reduced by the value $\vec{b}$. Therefore, $\vec{B}$ is a random value from $[-b, b]$, where the value of $b$ is decreasing from 2 to 0 during the iterations.

### 2. Spiral updating position

In spiral method, a relationship between the location of prey and whale to impersonate the helix-shaped operations is represented in the form of mathematical equation of humpback-whales in the following manner:

$$\vec{Y}(t) = \vec{E}' . e^{pq} . \cos(2\pi q) + \vec{X}^* \tag{21}$$

where $\vec{E}' = |\vec{X}^*(t) - \vec{X}(t)|$, which represents the distance between prey and the $i-th$ whale. $l$ denotes the random number $(-1 \leq l \leq 1).b$ represents logarithmic spiral, which is a constant number and . indicates the multiplication operation. It's worth noting that humpback-whales swim in a shrinking-circle around their prey while still following a spiral-shaped direction. To predict this concurrent action, an equation is derived to represent the model can be expressed as:

$$\vec{Y}(t) = \begin{cases} \vec{X}^*(t) - \vec{E} . \vec{B}, & \text{if } d < 0.5 \\ \vec{E}' . e^{pq} . \cos(2\pi q) + \vec{X}^*, & \text{if } d \geq 0.5 \end{cases} \tag{22}$$

where $d$ represents a random number $(0 \leq d \leq 1)$. Further, the searching behaviour of humpback-whales for prey is randomly in the bubble-net approach. The following is the representation of mathematical model for bubble net approach.

### Prey searching technique

To locate prey, same strategy based on the modification of the $\vec{B}$ vector can be utilized (exploration). In reality, humpback-whales search at random based on their location. As a result, we select $\vec{B}$ randomly with values $\vec{B} > 1$ or $\vec{B} < -1$ to compel the search-agent to step away from a target value. Comparison with exploitation, modify the location of every search-agent in the sample space, based on randomly selected process until to obtained a better solution. This operation and $|\vec{B}| > 1$ place an emphasis on the exploration phase and enable WOA to perform global-searching. This can be represented below:

$$\vec{E} = |\vec{A} . \vec{X_{rand}} - \vec{X}| \tag{23}$$

$$\vec{Y}(t) = \vec{X_{rand}} - \vec{B} . \vec{E} \tag{24}$$

where $\vec{X_{rand}}$ indicates a position-vector that is randomly selected from the existing space.

The algorithm WOA comprised of a selection of random samples. For every iteration, the search agents change their locations in relation to either a randomly selected search agent or the best solution acquired so far in this. For both cases exploitation phase and exploration the value of $b$ is decreasing in the range from 2 to 0 accordingly. As the value of $|\vec{B}| > 1$, a randomly searching solution is selected, while the optimal solution is obtained when $|\vec{B}| < 1$ for updating the search- location of the agents. Based on the parameter $d$, the WOA is used as a circular or spiral behaviour. Ultimately, the WOA is ended by the successful termination condition is met. Theoretically, it provides exploration and exploitation capability. Therefore, WOA can still be considered as a successful global optimizer. The WOA is described in Algorithm 1.

**Data**: Set the input control variables
$M, N, \gamma_m, \{g_m\}, \{s_n\}$
Population initialization $X_1, X_2, .........X_n$

**Result**: $X^*$ (Best search agent for user- pairing/grouping).

List all the users with decreasing order of $J_m$.
**while** $t < (total\ iterations)$
  **for** *every search user*
  Initialize b,B,A,q and d
    **if1**$(d < 0.5)$
      **if2**$(|B| < 1)$
      Existing search user position is updated using equation (16)
      **else if2**$(|B| \geq 1)$
      Randomly selected a search user $(X_{rand})$
      Existing position of search user is updated using equation (23)
      **end if2**
    **else if1**$(d \geq 0.5)$
    Exiting position of search user is updated using equation (21)
    **end if1**
  **end for**
  Examine the position of every search user in the search region if above the search region then modify it.
  Determine the position of every search user
  If a best solution becomes available, update X⋆
  t=t+1
**end while**
**return** $X^*$

**Algorithm 1:** WOA

## Grey wolf optimizer (GWO)

A popular meta-heuristic algorithm, which is influenced by the behaviour of grey-wolves (*Mirjalili, Mirjalili & Lewis, 2014*). This algorithm is based on the hunting approach of grey

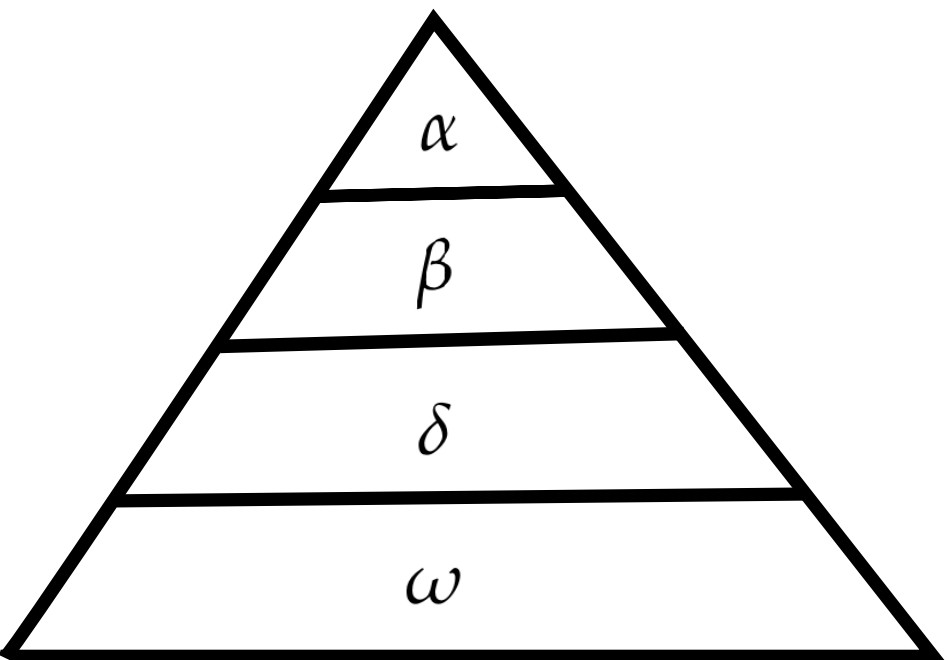

**Figure 2  Grey wolf hierarchy (*Mirjalili, Mirjalili & Lewis, 2014*).**

wolves and their governing- hierarchy. Grey wolves represent predatory animals, which means these are heading up in the hierarchy. Grey wolves tended to stay in groups. The wolves in a group is varying between 5 to 12. The governing-hierarchy of GWO is shown in Fig. 2, where several kinds of grey wolves have been used particularly $\alpha$, $\beta$, $\delta$, and $\omega$.

Both wolves (male and female) are the founders known as $\alpha$s. The $\alpha$ is mainly in favour of producing decision making regarding hunting, sleeping and waking time, sleeping place *etc.* The group is governed by the $\alpha$s actions. Even so, some egalitarian behaviour has been observed, such as an alpha wolf following other wolves in the group. The whole group respects the $\alpha$ by keeping their tails towards ground at gatherings. The $\alpha$ wolf is also regarded as superior since the group must obey his/her orders. The group's $\alpha$ wolves are the only ones that can mate. Usually, the $\alpha$ is not always the biggest member of the group, but rather the best at handling the batch. Which illustrates that a group's structure and discipline are often more critical than its capacity. $\beta$ is the second phase of the grey wolf hierarchy. The $\beta$'s are the sub-ordinate wolves who assist the $\alpha$ in taking decision. The $\beta$ wolf (male or female), is most likely the better choice to be the $\alpha$ wolf in the event that one of the $\alpha$ wolves dies or gets very old. Therefore, $\beta$ wolf would honour the $\alpha$ while still commanding all other lower-level wolves. It serves as an adviser to the $\alpha$ and a group disciplinarian. Throughout the group, the $\beta$ confirms the $\alpha$s orders and provides guidance to the $\alpha$. The grey wolf with the lowest rating is $\omega$. The $\omega$ serves as a scapegoat. $\omega$ wolves must still respond towards other dominant wolves in a group. They are the last wolves permitted to feed. While it might seem that the $\omega$ is not a vital member of the group, it has been found that when the $\omega$ is lost, the entire group experiences internal combat

and problems. This would be attributed to the $\omega$ venting his anger and resentment on both wolves (s). This tends to please the whole group while still preserving the dominance system. In certain circumstances, the $\omega$ is also the group's babysitter. Whenever a wolf should not be an $\alpha$, $\beta$, or $\omega$, he or she is referred to as a subordinate also called $\delta$. where $\delta$ wolves must yield to $\alpha$s and $\beta$s, however they rule the $\omega$. This group contains scouts, sentinels, elders, hunters, and caregivers. Scouts are in charge of patrolling the area and alerting the group if there is any threat. Sentinels defend and ensure the group's safety. Furthermore, the mathematical model of GWO is described as:

### Social hierarchy

For mathematical representation of GWO, we assume that $\alpha$ is fittest alternative solution used to mimic the social hierarchy. As a result, the second and third best solutions are designated as $\beta$ and $\delta$, respectively. The remaining member approaches are now considered to be $\omega$. For algorithm, the hunting (optimization) is led by $\alpha$, $\beta$ and $\delta$. These three wolves are accompanied by the $\omega$.

### Encircling prey

Grey wolves encircle prey during the hunting. The following equations are presented to mathematical model the encircling actions.

$$\vec{A} = |\vec{D}.\vec{X_s}(t) - \overrightarrow{X(t)}| \tag{25}$$

$$\vec{Y}(t) = \vec{X}(t+1) \tag{26}$$

$$\vec{Y}(t) = \vec{X_s}(t) - \vec{B}.\vec{A} \tag{27}$$

where $\vec{B}$ and $\vec{D}$ represents coefficient vectors, $t$ is the exiting iteration, $\vec{X}$ and $\vec{X_s}$ defines the position vector of a grey wolf and prey.

The vectors and are computed in the following manner:

$$\vec{B} = 2.\vec{b}.\vec{d_1} - \vec{b} \tag{28}$$

$$\vec{D} = 2.\vec{d_2} \tag{29}$$

where $0 \leq d_1 \leq 1$ and $0 \leq d_2 \leq 1$ indicates random vector and $\vec{b}$ component decreasing linearly over the entire iteration from 2 to 0.

### Hunting

Grey wolves do have capability to detect and encircle prey. The $\alpha$ normally leads the chase. The $\beta$ and $\delta$ can also engage in hunting. However, in an arbitrary search space, we have no idea that where is the optimal (prey) location. Hence, first acquired the three best solutions so far and then search other agents (containing $\omega$'s) in accordance with the best search agent's location. In this respect, the following equations are provided.

$$\vec{A_\alpha} = |\vec{D_1}.\vec{X_\alpha} - \vec{X}| \tag{30}$$

$$\vec{A_\beta} = |\vec{D_2}.\vec{X_\beta} - \vec{X}| \tag{31}$$

$$\vec{A_\delta} = |\vec{D_3}.\vec{X_\delta} - \vec{X}| \tag{32}$$

$$\vec{X_a} = \vec{X_\alpha} - \vec{\beta_1}.(\vec{A_\alpha}) \tag{33}$$

$$\vec{X_b} = \vec{X_\beta} - \vec{\beta_2}.(\vec{A_\beta}) \tag{34}$$

$$\vec{X_c} = \vec{X_\delta} - \vec{\beta_3}.(\vec{A_\delta}) \tag{35}$$

$$\vec{Y}(t) = \frac{\vec{X_a} + \vec{X_b} + \vec{X_c}}{3} \tag{36}$$

### *Attacking prey*
The wolves complete the chase by hitting the target once it cease running. The mathematical model of attacking prey approach can be achieve by decreasing the value of $\vec{b}$. It's worth mentioning that the variance range of $\vec{B}$ is also limited by $\vec{b}$.

That is $\vec{B}$ in the range of $[-b, b]$, which is a random value and decreasing over the entire iteration from 2 to 0. If any random values ranges between $[-1, 1]$, then new location of a search-agent lies between exiting and prey location.

### *Search for prey*
Grey wolves primarily hunt depending on the locations of the $\alpha$, $\beta$, and $\delta$. At the starting they diverge from other wolves to hunt and then combine to hit prey. The mathematical model of divergence can be achieved by utilizing the value of $\vec{B}$. For divergence, random values of $\vec{B} < 1$ or $\vec{B} > 1$ is used by the search agent. This process enables the GWO algorithm to search globally. In nature, the $D$ vector can even be assumed as the impact of barriers to pursuing prey. In general, natural barriers arise in wolves' hunting paths and discourage them from approaching prey effectively and easily. This is precisely depend on the vector $D$. It will arbitrarily give the prey a weight to find it tougher and farther to catch for wolves, depending on location of the wolf, or likewise.

The suggested social hierarchy supports GWO algorithm in sustaining the best solutions achieved so far through iteration. By using hunting approach, it enables agents to search the likely location of prey. The GWO is described in Algorithm 2.

**Data**: Set the input control variables
$M, N, \gamma_m, \{g_m\}, \{s_n\}$
Population initialization $X_1, X_2, \ldots\ldots X_n$
Initialization of $B, b$ and $D$

**Result**: $X_\alpha$ (Best search agent for user- pairing/grouping).

List all the users with decreasing order of $J_m$.
Determine fitness of every search agent
$X_\alpha, X_\beta$ and $X_\delta$.
**while** $t < (total\ iterations)$
   **for** $every\ search\ user$
      Update the existing location of search agent by using equation (36)
      Update $B$, $b$ and $D$
      Determine fitness of search agent
      $Update X_\alpha$, $X_\beta$ and $X_\delta$
      t=t+1
   **end for**
  **end while**
  **return** $X_\alpha$

**Algorithm 2:** GWO

## Particle swarm optimization (PSO)

*Kennedy & Eberhart (1995)* introduced PSO as an evolutionary computation method. It was influenced by the social behaviour of birds, which involves a large number of individuals (particles) moving through the search space to try to find a solution. Over the entire iterations, the particles map the best solution (best location) in their tracks. In essence, particles are guided by their own best positions, which is the best solution same as achieved by the swarm. This behaviour can modelled mathematically by using velocity vector $(u)$, dimension $(S)$, which represents the number of parameters and position vector $(x)$. In the entire iterations, the position and velocity of the particles changing by the following equation:

$$u_i^{t+1} = vu_i^t + e_1 \times rand \times (pbest_i - x_i^t) + e_2 \times rand \times (gbest - x_i^t) \tag{37}$$

$$x_i^{t+1} = u_i^{t+1} + x_i^t \tag{38}$$

where $v(0.4 \leq v \leq 0.9)$ represents the inertial weight, which control stability of the PSO algorithm. cognitive coefficient $e_1(0 < e_1 \leq 2)$, which limits the impact of the individual memory for best solution. Social factor $e_2(0 < e_2 \leq 2)$, which limits the motion of particles to find best solution by the entire swarm, *rand* indicates a random number in the range between 0 and 1, attempt to provide additional randomized search capability to the PSO algorithm and two variables *pbest* and *gbest*, used to accumulate best solutions achieved by each particle and the entire swarm accordingly. The PSO is described in Algorithm 3.

**Data**: Set the input control variables
$M, N, \gamma_m, \{g_m\}, \{s_n\}$
Population initialization $X_1, X_2, \ldots \ldots X_n$

**Result**: *pbest and gbest* (Best search agent for user- pairing/grouping).

List all the users with decreasing order of $J_m$.
 **for** *each generation* **do**
  **for** *each particle* **do**
  Update the position and vector by using equation (37) and equation (38)
Estimate the fitness of the particle
  Update both *pbest* and *gbest*
  t=t+1
  **end for**
 **end for**
**return** *pbest*, *gbest*

**Algorithm 3:** PSO

**Table 1  Parameters for proposed uplink NOMA.**

| Parameter | Value |
| --- | --- |
| $C$ | 1 |
| $M$ | 6 |
| $N$ | 3 |
| $s_m$ | 1.1 bits/s/Hz |
| $\gamma$ | 30 dB |

## SIMULATION RESULTS

This section evaluates the performance of the proposed meta-heuristic algorithms, namely, WOA, GWO and PSO for joint problem of user-grouping, power control and decoding order for NOMA uplink systems.

Table 1 presents the simulation parameter values attained from the literature (*Sedaghat & Müller, 2018*; *Zhang et al., 2019*; *Mirjalili & Lewis, 2016*; *Mirjalili, Mirjalili & Lewis, 2014*; *Kennedy & Eberhart, 1995*) for WOA, GWO and PSO algorithms that involved in the simulation. Further, the Wilcoxon test and Friedman test (Abualigah et al. (2921)) are performed for experiments and the statistical analysis of GWO and PSO is also provided in Table 2. Based on the results of tests, the proposed WOA outperforms the other algorithms in comparison.

Both channel of the users and location are allocated randomly in the simulation. Therefore, the range between the user and BS are uniformly distributed and considered that the channel response is Gaussian distribution (*Zhang et al., 2019*).

Figure 3 indicates the comparison of convergence of WOA (*Mirjalili & Lewis, 2016*) , GWO (*Mirjalili, Mirjalili & Lewis, 2014*) and PSO (*Kennedy & Eberhart, 1995*) algorithms proposed for NOMA uplink system. We may conclude that WOA, GWO and PSO algorithms converge at a comparable rate, hence WOA converges after a greater number

**Table 2** Statistical analysis of GWO and PSO.

| Wilcoxon | GWO | PSO |
|---|---|---|
| *p*-value | $1.8E - 169$ | $4.7E - 181$ |

| Friedman | GWO | PSO |
|---|---|---|
| *p*-value | $4.5E - 161$ | $1.2E - 164$ |

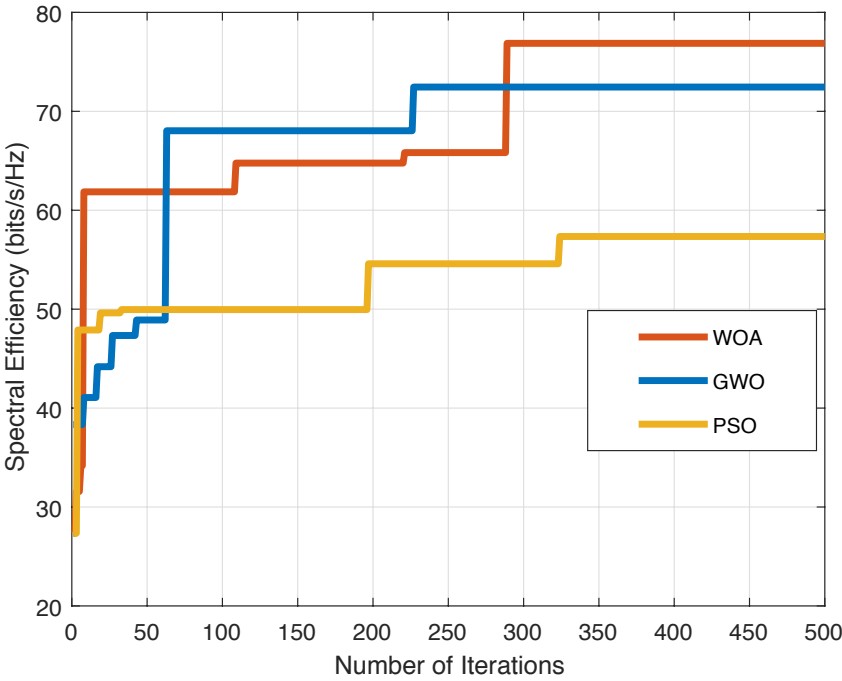

**Figure 3** Illustration of convergence of WOA, GWO and PSO.

of iterations than GWO and PSO. The proposed WOA attains significant performance in-terms of spectral efficiency as compare to GWO and PSO algorithms. The proposed WOA (*Mirjalili & Lewis, 2016*) provides stability and attains the minimum rate requirement without such a noticeable drop in the results.

Figure 4 compares the spectral-efficiency of NOMA and OMA approaches with varying $\gamma$, respectively. It has been proved that the spectral-efficiency of NOMA scheme is considerably higher than those of scheme. Moreover, the spectral-efficiency of the proposed sub-optimal approach is nearer to the optimal value. The proposed WOA algorithm attains near optimal performance with minimal computational complexity. In addition, as the number of users increases the computational cost of the exhaustive-search algorithm increases as compared to WOA.

For NOMA uplink systems, the power control approach in *Sedaghat & Müller (2018)* is provided as a benchmark scheme, where the spectral efficiency are near to the optimal value. Noted that the approach used in *Sedaghat & Müller (2018)* is valid only for two

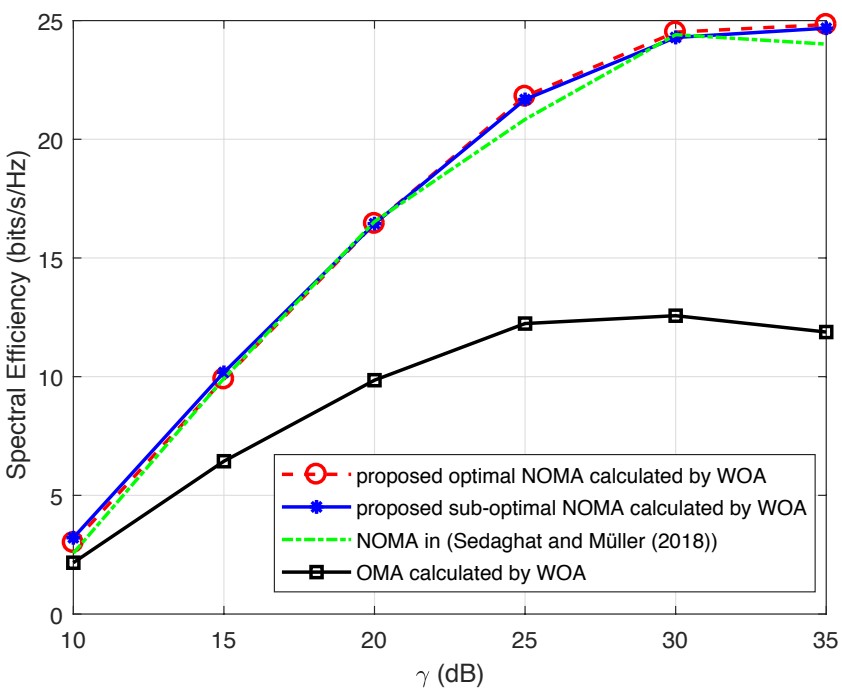

**Figure 4** **Illustration of spectral efficiency of WOA with increasing $\gamma$.**

user-pairing. Hence, the proposed scheme performs admirably in-terms of having efficient user grouping for multiple users.

Figures 5 and 6 evaluates the performance of GWO and PSO algorithms in-terms of spectral-efficiency. For uplink NOMA system, the spectral-efficiency of NOMA scheme outperform OMA scheme with varying $\gamma$. Moreover, the spectral-efficiency of optimal and sub-optimal solutions are nearer to each other. The power control scheme for NOMA uplink system in *Sedaghat & Müller (2018)* is used as a benchmark. It has been observed that the spectral-efficiency of both GWO and PSO algorithms shows better results than power control (*Sedaghat & Müller, 2018*) and OMA scheme.

Moreover, a comparison of proposed optimal WOA, GWO and PSO has shown in Fig. 7. The performance of proposed optimal WOA, GWO and PSO are almost nearer to one another. Moreover, as the value of $\gamma$ above 30 dB, the optimal WOA performs better in-terms of spectral-efficiency as compare to GWO and PSO.

## CONCLUSION

NOMA systems have garnered a lot of interest in recent years for 5G cellular communication networks. The efficient user grouping and power control scheme play an essential role to enhance the performance of communication network. In this paper, we have examined for the first time up to authors knowledge, a joint issue of user-grouping and power control for NOMA uplink systems. we have solved this problem by proposing WOA with low complexity. Further, for comparison, GWO and PSO were adopted to solve the

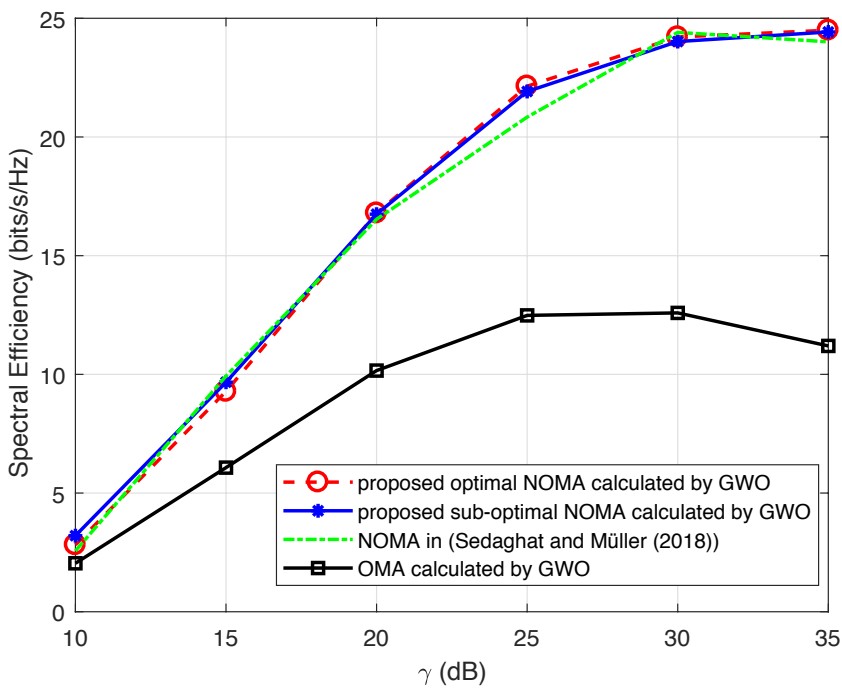

**Figure 5** **Illustration of spectral efficiency of GWO with increasing γ.**

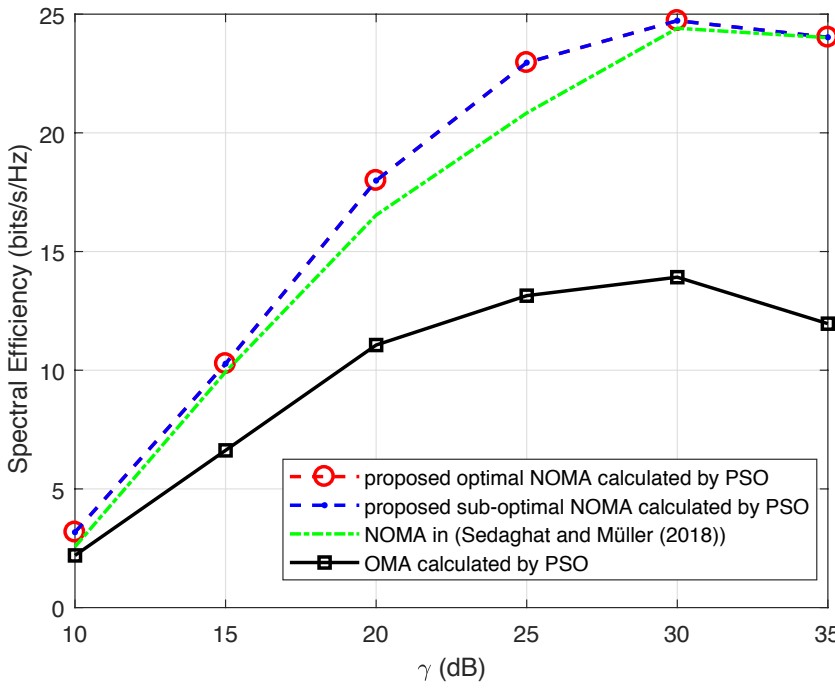

**Figure 6** **Illustration of spectral efficiency of PSO with increasing γ.**

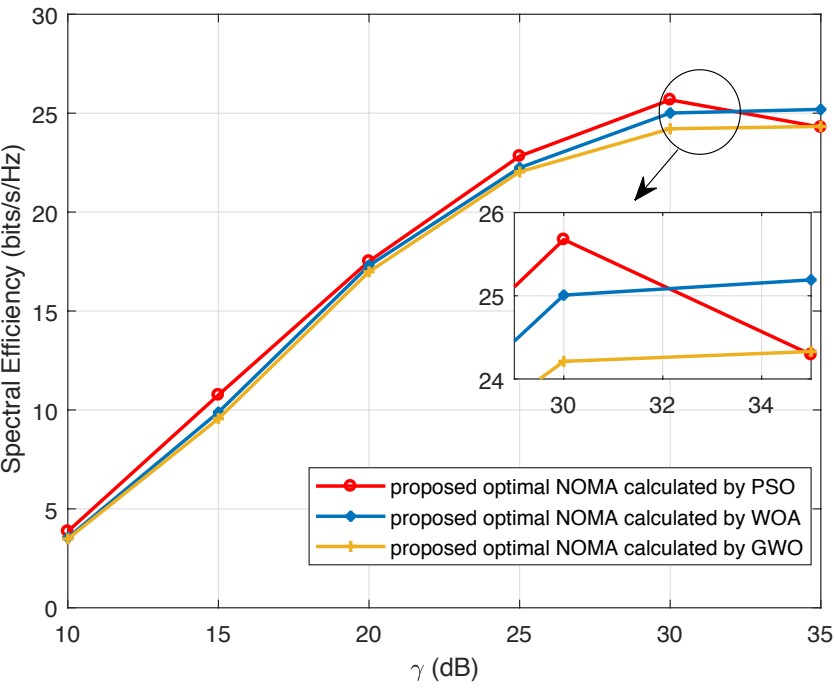

**Figure 7** Comparison of optimal solution of WOA, GWO and PSO.

same problem. Simulations results show that the WOA proposed for this combine issue in uplink allows better performance than the conventional OMA in-terms of spectral-efficiency. Further, proposed WOA provides better result as compared to GWO, PSO and existing algorithm in literature with lower system complexity by considering same constraint regarding uplink NOMA systems.The acquired results also suggest that the combinatorial joint problem gets more difficult to solve as the number of users grows and needs additional network resources. In the future, the study might be expanded to include more performance parameters to the mentioned problem and implementation of multiple antennas combinations which leads to massive MIMO (Mulitiple-Input and Multiple-Output) scenario in order to further enhance the performance of the network.

### Funding
The authors received no funding for this work.

### Competing Interests
The authors declare there are no competing interests.
## Author Contributions

- Bilal ur Rehman conceived and designed the experiments, performed the experiments, analyzed the data, performed the computation work, prepared figures and/or tables, authored or reviewed drafts of the paper, and approved the final draft.
- Mohammad Inayatullah Babar conceived and designed the experiments, performed the experiments, analyzed the data, prepared figures and/or tables, authored or reviewed drafts of the paper, and approved the final draft.
- Arbab Waheed Ahmad, Muhammad Amir and Waleed Shahjehan conceived and designed the experiments, performed the experiments, analyzed the data, prepared figures and/or tables, and approved the final draft.
- Ali Safaa Sadiq performed the computation work, prepared figures and/or tables, authored or reviewed drafts of the paper, and approved the final draft.
- Seyedali Mirjalili and Amin Abdollahi Dehkordi performed the computation work, prepared figures and/or tables, and approved the final draft.

## Data Availability

The code is available in the Supplemental Files.

## Supplemental Information

Supplemental information for this article can be found online at http://dx.doi.org/10.7717/peerj-cs.882#supplemental-information.

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
