# Peer review of "Joint user grouping and power control using whale optimization algorithm for NOMA uplink systems"

_PeerJ Computer Science, doi:10.7717/peerj-cs.882_

## Round 0.1 · original submission · Major Revisions

As pointed out by the reviewers, the contributions of the work should be clearly discussed in the revised draft.

Reviewer 1 ·

Basic reporting

"The authors introduced Whale Optimization Algorithm (WOA) and compare it with Grey Wolf Optimizer (GWO) and Particle Swarm Optimization (PSO) algorithms in the context of NOMA-based wireless system. The manuscript is easy to follow. The authors can consider the below suggestions to improve the quality of the paper.

As for the writing:
- abstract: "A user grouping,... assume..." to passive voice.
- please avoid using ambiguous expressions such as "high", "significantly comparable", etc.
- line 144 two "at".
- line 151 unfinished sentence.
- avoid starting sentences with Greek symbols or "Which".
- Grey Wolf Optimizer (GWO): The gender of the wolve is emphasized. How does it contribute to the algorithm?
- "Figure 2: Grey wolf hierarchy ?"
- line 379 "is assume" to passive voice.
- line 412 "simulations results"

Experimental design

The authors proper explanation NOMA-based wireless system performance and detailed analysis. Neither experimental verification nor comparison to other methods/models is presented. and more comparison to recently published papers that may prove scientific value is not provided

Validity of the findings

Please restructure the paper, as sections "Spectral efficiency maximization" and "Solution of proposed model" can be subsections instead of being equal to sections "Introduction" and "Discussion".

If possible, please elaborate the discussion on the Simulation results and add application scenarios and/or future works in the Conclusion.

Reviewer 2 ·

Basic reporting

no comment

Experimental design

no comment

Validity of the findings

no comment

Additional comments

This paper studies the uplink NOMA system with the WOA algorithm, But the reviewer cannot accept this manuscript. Comments are as follows:
1. The authors just applied the WOA algorithm to the uplink NOMA systems and the reviewer could not find any interesting idea or intuition from this manuscript.
2. No performance gain of the WOA algorithm compare with PSO or GWO algorithm from fig.7

3. Future works should be added in the last section.

Reviewer 3 ·

Basic reporting

Although the language of the article is understandable, it needs some enhancements, for example, when the authors wrote "In this work, a newly developed Whale Optimization Algorithm (WOA) is implemented", they did not mention the purpose of using this algorithm specifically. And the article also lacks clarity when talking briefly about results in the Abstract section as in the sentence " Also, WOA attains improved results in compliance with system complexity". Moreover there are some grammatical mistakes and types like: "is capable to chose", "would honour the","where as ", "response is assume to be", "are describe as"....

Experimental design

WOA algorithm has to be written and described by taking into account the NOMA Uplink system,
One of the drawbacks of the article is that the baselines contain more details than the proposed algorithms.

Validity of the findings

I think it would be good to compare your results against other algorithms such as Cuckoo Search Optimization, Ant Colony Algorithm, and other state-of-the-art algorithms...

Additional comments

The article is missing the future work

Reviewer 4 ·

Basic reporting

The problem under study is interesting and trendy, but the major issue is that there is no significant and convincing novelty. In the following, some concerns will be discussed which hopefully can help the authors.

1. The main concern about this work is that the manuscript consists of contradictory statements about the contribution. It seems the authors investigated the usage of three algorithms on the application of NOMA, not proposing a new variant of WOA. The authors should specify with clarity their novelty and contribution. For instance, the reader encountered the following inconsistent sentences,
- “In this work PSO, GWO, and WOA are employed”, “a newly developed WOA is implemented”, “To solve the issue of complexity, a WOA with low complexity is investigated for an efficient opt…”, “The proposed algorithm for user pairing/grouping”, “… reduce the system complexity, an innovative existence metaheuristic optimization technique named WOA is proposed in this work.”, “This section evaluates the proposed algorithms WOA, GWO, and PSO…”.
2. The abstract should be revised to reflect the main novelty and contribution of the paper.
3. The literature overview in Introduction section is shallow, the authors should add an in-depth literature review of recent optimization algorithms such as MTDE and I-GWO (with DLH search strategy) algorithms.
4. It is recommended to provide more informative literature on the usage of metaheuristic algorithms for solving the NOMA uplink system.
5. The Section entitled “Optimal and Sub-optimal User Grouping using WOA” is vague and does not show the methodology of the usage of WOA for user grouping tasks. The authors should describe the proposed algorithm step-by-step and show the sequence by using a flowchart or pseudo-code. (Algorithm 1 is the pseudo-code of WOA not the usage of it for user grouping)
6. The authors should clarify what is the “WOA NOMA”. There is no explanation for “WOA NOMA” in the manuscript until the Simulation results section.

Experimental design

7. The authors mentioned in Introduction that “The results obtained through the algorithms proposed in (Sedaghat and Mu¨ ller (2018)), WOA, GWO and the popular PSO are exclusively compared in this study.”, but there are not any results from (Sedaghat and Mu¨ ller (2018) in the Simulation results section.
8. The authors should correct the legend of Figures 4-6 to remove “[19]” and add a proper reference.
9. The selection of comparative algorithms is not satisfactory. It’s necessary to compare the proposed algorithm with recently proposed and enhanced variants of algorithms.
10. I could not find the number of iterations, runs, and population used in experiments; are they missed?

Validity of the findings

11. The authors should statistically analyze the proposed and comparative algorithms using Wilcoxon and Friedman tests.
12. The authors have claimed about the aim of using WOA and “obtaining high spectral-efficiency with lower computational complexity”, but this complexity is related to WOA and the authors have no contribution to achieve or even to reduce it.
13. It is recommended to analyze the performance and effectiveness of the proposed and comparative algorithms using the performance index (PI) as shown in Subsection 5.3.5 in the QANA paper.

Additional comments

14. It is noted that Figures 4, 5, 6, and 7 are illustrated before their description.

---

## Round 0.2 · Minor Revisions

Please address the comments and resubmit. The reviewer (#3) has requested that you cite specific references. You may add them if you believe they are especially relevant. However, I do not expect you to include these citations, and if you do not include them, this will not influence my decision.

Reviewer 3 ·

Basic reporting

The article appears much better. The content is clear and easy to understand.
However, I recommend referring to some recent research papers that are considered as related to this research paper:
- Alawad, N. A., & Abed-alguni, B. H. (2021). Discrete Island-Based Cuckoo Search with Highly
Disruptive Polynomial Mutation and Opposition-Based Learning Strategy for Scheduling of
Workflow Applications in Cloud Environments. Arabian Journal for Science and Engineering, 46(4),
3213-3233.
- Panda, S. (2020). Joint user patterning and power control optimization of MIMO–NOMA systems. Wireless Personal Communications, 1-17.
- Zheng, G., Xu, C., & Tang, L. (2020, May). Joint User Association and Resource Allocation for NOMA-Based MEC: A Matching-Coalition Approach. In 2020 IEEE Wireless Communications and Networking Conference (WCNC) (pp. 1-6). IEEE.
- Abed-alguni, B. H., & Alawad, N. A. (2021). Distributed Grey Wolf Optimizer for scheduling of
workflow applications in cloud environments. Applied Soft Computing, 102, 107113

Experimental design

The methods and figures are described snd formulated well.

Validity of the findings

No comment.

Additional comments

No comment.

---

## Round 0.3 · accepted · Accept

All concerns have been addressed. Congratulations

Reviewer 3 ·

Basic reporting

no comment

Experimental design

no comment

Validity of the findings

no comment

Additional comments

no comment.
Everything has been addressed well.